# Gait Domains May Be Used as an Auxiliary Diagnostic Index for Alzheimer’s Disease

**DOI:** 10.3390/brainsci13111599

**Published:** 2023-11-17

**Authors:** Qi Duan, Yinuo Zhang, Weihao Zhuang, Wenlong Li, Jincai He, Zhen Wang, Haoran Cheng

**Affiliations:** 1Department of Neurology, The First Affiliated Hospital of Wenzhou Medical University, Wenzhou 325000, China; qiduan0314@163.com (Q.D.); zwh19980816@163.com (W.Z.); hjc@wmu.edu.cn (J.H.); wangzhen@wzhospital.cn (Z.W.); 2Department of Psychiatry, Wenzhou Seventh People’s Hospital, Wenzhou 325000, China; yinuo_zhang818@163.com; 3Radiotherapy Center, The First Affiliated Hospital of Wenzhou Medical University, Wenzhou 325000, China; lwl202310@163.com

**Keywords:** Alzheimer’s disease, gait domain, principal components factor analysis, dual-task gait assessments

## Abstract

Background: Alzheimer’s disease (AD) is a progressive neurodegenerative disorder with cognitive dysfunction and behavioral impairment. We aimed to use principal components factor analysis to explore the association between gait domains and AD under single and dual-task gait assessments. Methods: A total of 41 AD participants and 41 healthy control (HC) participants were enrolled in our study. Gait parameters were measured using the JiBuEn^®^ gait analysis system. The principal component method was used to conduct an orthogonal maximum variance rotation factor analysis of quantitative gait parameters. Multiple logistic regression was used to adjust for potential confounding or risk factors. Results: Based on the factor analysis, three domains of gait performance were identified both in the free walk and counting backward assessments: “rhythm” domain, “pace” domain and “variability” domain. Compared with HC, we found that the pace factor was independently associated with AD in two gait assessments; the variability factor was independently associated with AD only in the counting backwards assessment; and a statistical difference still remained after adjusting for age, sex and education levels. Conclusions: Our findings indicate that gait domains may be used as an auxiliary diagnostic index for Alzheimer’s disease.

## 1. Introduction

Alzheimer’s disease (AD) is a progressive neurodegenerative disorder with cognitive dysfunction and behavioral impairment. It is estimated that as of 2019, there were approximately 50 million people with dementia worldwide, which may increase to 152 million by 2050 [1]. AD is the most common cause of dementia and accounts for 60–80% of all dementia cases. The diagnostic criteria of AD incorporate AD biomarkers based on magnetic resonance imaging (MRI), positron emission tomography (PET) imaging and cerebrospinal fluid (CSF) assays [2]. However, the AD biomarker tests based on PET neuroimaging and CSF assays are invasive, expensive and inconvenient to use. Meanwhile, the amount of commonly used AD drugs is limited, and these drugs temporarily relieve symptoms, slow down the disease progression and cannot reverse the course of the disease [3]. The prevalence of AD has imposed a huge economic burden on families and society [4]. Hence, early diagnosis of AD is critical for its treatment, management and prognosis.

Gait is a complex motor task; normal gait is regulated by multiple systems, such as the nervous system, musculoskeletal system and cardiorespiratory systems [5]. Unlike cognitive tests, gait assessment is a common component of physical examinations across a variety of disciplines. Diseases affecting each of these systems which are involved in gait may cause gait impairment. Meanwhile, aging and neuropathological changes have a detrimental effect on postural control and lead to gait disturbance. Compared with young people, reduced gait velocity and step length and an increased double stance time are observed in aging individuals due to structural and functional brain changes and muscle atrophy as previously reported [6,7]. In addition, gait disturbances are common findings in Parkinson’s disease (PD), manifesting as reduced speed, shorter step length, increased stride-to-stride variability, reduced automaticity, increased gait asymmetry and freezing of gait (FOG) [8,9]. Additionally, Al-Yahya et al. found that damage to the prefrontal cortex due to stroke is related to gait impairment, especially dual-task walking [10]. In recent years, accumulating evidence suggests that gait disturbance is associated with cognitive impairment [11,12,13,14]. Cognitive functions, especially executive function and working memory, play an important role in the regulation of gait. A previous study found that declines in executive functions and working memory were associated with a decline in gait velocity in older adults with mild cognitive impairment (MCI) [11]. Furthermore, high stride-to-stride variability of stride time (STV) at a fast-pace walking speed was proven to be a specific gait disturbance of MCI patients [12]. The dual-task paradigm, performing different cognitive tasks while walking, has been used to assess the interactions between gait and cognition [13,14]. The dual-task paradigm affords the opportunity to manipulate attention demands. Thus, the dual-task paradigm may reflect the relationship between cognition and gait more sensitively compared with the single-task paradigm.

Artificial intelligence (AI) has led to numerous technical innovations in medicine and revolutionized the conventional mode of medicine, especially neurological diseases. For example, using the method of human–computer interaction for early warning and ancillary diagnosis of nervous system diseases [15]. The brain—computer interface (BCI) is the linkage of the brain to computers through scalp, subdural or intracortical electrodes to improve control of movement disorders and memory enhancement [16]. Sensor technology is the most basic accessory of artificial intelligence. Wearable intelligent sensors are inexpensive, convenient and efficient, which has made them one of the most popular types of electrochemical sensors. Meanwhile as a clinical tool applied in the rehabilitation and diagnosis of medical conditions and sport activities, gait analysis using wearable sensors shows great prospects [17]. By means of this technology, several studies have demonstrated that a daily poor gait performance is associated with the risk of falls [18], Parkinson’s disease [8] and AD [19].

However, gait parameters are closely correlated with each other, and gait parameters alone may not fully explain the gait performance. Recently, the concept of gait domains is widely used in gait performance of different diseases or life stages. In 2011, Hollman et al. identified five gait domains including rhythm, phases, variability, pace and base of support in old people, based on 22 individual gait parameters [20]. A prospective cohort study of 427 old people found that the pace factor could predict the risk of cognitive decline and developing vascular dementia in the early stage [21]. Attentional control and pace factors were found to heavily influence the relationship of cognitive function and gait in early PD [22]. Considering the effects of levodopa on PD, a previous study explored the gait domains in the OFF and ON levodopa states and found that the variability factor was greater in the OFF medication state [23]. 

Given that the gait parameters are closely correlated with each other, and gait parameters alone may not fully explain the gait performance, we used a factor analysis to identify independent gait domains derived from quantitative assessments to address this issue. A principal components factor analysis organizes multiple observations into communalities that correlate with a lesser number of unobserved thematic constructs, thus allowing an investigator to partition a large number of parameters into a lesser number that characterize distinct domains of the parameters being measured [24].

Therefore, the purpose of this study was twofold. Firstly, the main purpose of this study is to explore the association between gait parameters and AD. Second, we sought to contribute to finding a new, objective and simple gait domain to recognize the AD patients from healthy old people in clinical practice. The gait domains could help clinicians to fully comprehend the gait performance in AD patients and help clinicians to choose the appropriate combination of gait parameters to monitor the disease and its progression.

## 2. Materials and Methods

### 2.1. Subjects

This study was a case–control study of AD participants and healthy control (HC) participants and was carried out at the Neurology Department of First Affiliated Hospital of Wenzhou Medical University. Collection of the data was continued from October 2019 to November 2021. This study was approved by the Ethics Committee of First Affiliated Hospital of Wenzhou Medical University (approval code: KY2021-153.) and all participants signed written informed consent prior to participation in accordance with the Declaration of Helsinki.

AD participants and HC participants were enrolled from the Memory Clinic of Neurology Department and the Health Examination Center of First Affiliated Hospital of Wenzhou Medical University, respectively. The inclusion criteria were as follows: (a) able to walk ten meters safely without assistance or auxiliary equipment; and (b) able to understand walking test instructions and complete them. The exclusion criteria were as follows: (a) dyskinesia or leg problems, such as knee replacements or hip replacements; (b) major central nervous system disease, such as stroke, Parkinson’s disease, Huntington’s disease or myasthenia gravis; (c) major psychiatric disorders which may impair cognition and gait, such as schizophrenia, bipolar affective disorder or alcohol abuse; (d) severely impaired cognitive function or unable to understand and complete the three prescribed walking tests; and (e) unwilling to sign the informed consent. A total of 41 AD participants and 41 normal cognitive HC participants were enrolled in our study according to our inclusion and exclusion criteria. 

Diagnosis of AD was according to the National Institute on Aging-Alzheimer’s Association (NIA-AA) [25,26] and Statistical Manual of Mental Disorders V (DSM-V) [27]. In the current study, AD patients had a duration range of 3–6 years, and all magnetic resonance imaging (MRI) results were documented. Based on clinical symptoms, MMSE scores and MRI, AD patients were diagnosed by a trained neurologist who was a specialist in dementia. HC participants were enrolled by the above inclusion and exclusion criteria. HC participants had intact cognitive function, and did not meet the criteria of dementia or MCI. MCI is a clinical diagnosis based on subjective cognitive decline, objective cognitive impairment and relative preservation of activities of daily living, which was diagnosed based on the criteria defined by Petersen et al. [28]. 

### 2.2. Demographic and Clinical Data Collection

We collected the participants’ demographic and clinical data such as age, gender, body mass index (BMI), educational information and medications. All participants underwent the Mini-Mental Status Examination (MMSE) to assess their cognitive functions on admission. MMSE was carried out in Chinese and by a trained neurologist who was blinded to the gait performances of the participants. Participants were considered to have cognitive impairment according to their education levels: when they were illiterate, MMSE scores ≤ 19 points; when they had primary school education, MMSE scores ≤ 22 points; when they had middle school education or and higher, MMSE scores ≤ 26 points [29]. Meanwhile, we assessed the use of medication to treat AD (aricept and/or memantine), depressive disorders (sertraline) and psychotic disorders (olanzapine, risperidone, quetiapine and cozapine), because these medications affect gait function [30].

### 2.3. Gait Assessment

Gait performance was measured using the JiBuEn^®^ gait analysis system (Hangzhou Zhihui Health Management Co., Ltd., Hangzhou, China) consisting of 5 inertial sensors and a pair of shoes with gyroscope and 32 pressure sensors [31]. Five inertial sensors are placed on the subject’s waist, thighs and calves using nylon straps. Signals from the sensors are sampled and transferred through Bluetooth and received by a receiver connected to a computer. JiBuEn^®^ system is one of the portable wearable devices that have been developed and used in measuring gait with low cost, simple implementation and even instant reporting [32]. The detailed experimental design, algorithm for gait parameters and validated method were reported in a previous study [33], and systematically evaluated the validity of JiBuEn^®^ [34]. The high-order low-pass filter and hexahedral calibration technique were employed in data pre-processing, which reduces high-frequency noise interference and installation errors produced by sensor devices. Moreover, accumulative errors were also corrected based on the zero-correction algorithm. The final gait parameters were obtained by fusing acceleration data and posture, which is calculated using quaternary complementary filtering technique. This gait system could record the detailed data of the walking process in a multi-dimensional manner and automatically calculate gait parameters in real time, such as gait velocity, length, stride time and gait variability. The gait cycle was defined as the process of walking with the same foot stepping off the ground from the heel to landing on the heel again (Figure 1). 

Gait assessments were held in a quiet well-lit environment and participants were well protected. Participants were asked to walk 10 m with different instructions in a fixed place where stop and start points had been marked on the floor. In our study, all participants performed two walk trials including free walk and counting backwards. The free walk was performed as follows: participants walked in their normal and daily walking state. The counting backwards was performed as follows: participants were asked to walk while counting aloud backward from 100 to 0 [35]. 

### 2.4. Statistical Analyses

For continuous variables, mean ± SD or medians (quartiles) were used for statistical descriptions; the Student’s *t*-test or the Mann–Whitney test was used to compare the intergroup differences. For categorical variables, relative frequencies and percentages (%) were used for statistical descriptions; the Chi-square test or Fisher’s exact test was used to compare the intergroup differences. 

Gait parameters were closely correlated with each other, and independent effects were hard to distinguish. In order to derive independent factors named gait domains, the principal component method was used to conduct an orthogonal maximum variance rotation factor analysis of quantitative gait parameters [24]. The initial factors were then subjected to an orthogonal varimax rotation to reduce the larger number of highly correlated variables to a smaller number of uncorrelated independent predictors to be used in the final analysis. Principal components factor analysis with varimax rotation was used to examine factors with eigenvalues exceeding 1.0 that characterized gait performance. Parameters with correlation loadings of 0.5 or higher were interpreted as being significant contributors to the factor. To assess the association between AD and gait domain for two gait assessments, the multivariate logistic regression model with a stepwise backward selection process was applied to adjust potential confounding or risk factors based on age, gender and education level [36]. Meanwhile, a forest plot has been developed to show the results of logistic regression. All statistical analyses were carried out using IBM SPSS Statistics for Windows, version 26.0 (IBM Corp, Armonk, NY, USA) and R software (https://www.r-project.org/, accessed on 1 December 2021 version 4.3.1). Two-tailed *p*-values less than 0.05 (*p* < 0.05) were considered statistically significant.

## 3. Results

### 3.1. Baseline Demographic and Clinical Characteristics

A total of 82 participants (41 AD and 41 HC) were enrolled in our study. The baseline demographics and clinical characteristics of the study are presented in Table 1. There was no significant difference in sex and BMI between the two groups. The mean age was 68.2 ± 8.1 years in the AD group; 26 (63.4%) participants were female; 12 (29.3%) participants were illiterate, 14 (34.1%) participants had a primary school education, 15 (36.6%) had a middle school education and higher; and MMSE scores were 13.0 (7.0–19.0). For clinical reasons, the elderly HCs were difficult to enroll. After adjusting for age, the MMSE scores of AD participants were still lower than for HCs (*p* = 0.002).

### 3.2. Gait Parameters

The gait parameters of the free walk and counting backwards are presented in Table 1. Compared with HCs, AD participants had a shorter stride length (free walk: *p* < 0.001, counting backwards: *p* < 0.001; Table 1), slower gait velocity (free walk: *p* = 0.001, counting backwards: *p* < 0.001; Table 1) and bigger proportion of the stance phase (free walk: *p* = 0.001, counting backwards: *p* < 0.001; Table 1), which were found in both the free walk and counting backwards. However, AD participants had a larger variability in stride time (*p* = 0.023; Table 1) and the swing phase (*p* = 0.004; Table 1), which were only found in counting backwards. Considering the influence of age, we repeated the multiple logistic regression to adjust for gender, age and education. Consistent with the above results, a shorter stride length (free walk: *p* = 0.006, counting backwards: *p* = 0.003; Table 2), slower gait velocity (free walk: *p* = 0.016, counting backwards: *p* = 0.007; Table 2) and bigger proportion of the stance phase (free walk: *p* = 0.024, counting backwards: *p* = 0.014; Table 2) were significantly associated with AD, which were found in both the free walk and counting backwards. A larger variability in stride time (*p* = 0.031; Table 2) and the swing phase (*p* = 0.040; Table 2) were significantly associated with AD, which were only found in counting backwards.

### 3.3. Gait Domains

Consistent with the previous research, the identified gait domains were derived from similar gait parameters [15]. In two gait assessments, the components of three gait domains were the same. The first factor was named the rhythm factor, loading heavily on the gait frequency, stance phase and swing phase. The second factor was named the pace factor, loading heavily on the stride length and gait velocity. The third factor was named the variability factor, loading heavily on the stride time variability and swing phase variability. The free walk yielded the gait domains as follows: rhythm factor (34.98%), pace factor (31.13%) and variability factor (28.46%). Counting backwards yielded the gait domains as follows: rhythm factor (40.23%), pace factor (27.86%) and variability factor (26.73%). The components and loadings mentioned above are presented in Table 3.

The multivariate regression analysis showed that the pace factor was an independent variable for recognizing AD in the two gait assessments (free walk: odds ratio [OR] = 0.324, 95% confidence interval [CI] = 0.180–0.583, *p* < 0.001; counting backwards: OR = 0.285, 95% CI = 0.151–0.540, *p* < 0.001; Figure 2); the variability factor was an independent variable for recognizing AD in counting backwards (OR = 2.575, 95% CI = 1.097–6.041, *p* = 0.030; Figure 2). After adjusting for age, sex and education levels, the multivariate regression analysis showed that the pace factor (OR = 0.405, 95% CI = 0.215–0.763, *p* = 0.005; Figure 2) was independently associated with AD in the free walk. For counting backwards, the pace factor (OR = 0.337, 95% CI = 0.169–0.674, *p* = 0.002; Figure 2) and variability factor (OR = 2.750, 95% CI = 1.115–6.781, *p* = 0.028; Figure 2) were significantly associated with AD.

## 4. Discussion

To the best of our knowledge, this is the first study to explore the association between gait domains and AD participants. Compared with HC, we found that AD participants had a shorter stride length, slower gait velocity and bigger proportion of the stance phase both in the free walk and counting backwards; there was a larger variability in the stride time and swing phase only in counting backwards, compared with HC. In two gait assessments, the components of three gait domains (rhythm, pace and variability domains) were found to be the same. The rhythm factor loaded heavily on the gait frequency, stance phase and swing phase; the pace factor loaded heavily on the stride length and gait velocity; and the variability factor loaded heavily on the stride time variability and swing phase variability. We found that the pace factor was independently associated with AD both in the free walk and counting backwards assessments; the variability factor was independently associated with AD only in the counting backwards assessment; and the statistical difference still remained after adjusting for age, sex and education levels. Gait domains could help clinicians to better understand the gait performance of AD participants and the relationships between several gait parameters, which would be helpful for selecting the appropriate gait domains for monitoring under single and dual-task walking conditions in AD participants. Therefore, gait domains (pace and variability domains) may be helpful for detecting AD in clinical practice. 

Our study indicated that the pace domain was highly associated with AD both in the single- and dual-task conditions. After adjusting for age, sex and education levels, a 1-point decline in the pace factor (free walk, by 59.5%; counting backwards, by 66.3%) was associated with the risk of developing AD. Cullen et al. conducted research on subjective cognitive impairment (SCI), MCI and dementia patients and found that dementia patients showed a slower gait speed than the other two groups, but there were no differences between the SCI and MCI groups [37]. Previous research suggested that reduced executive function and working memory performances were associated with slow gait velocity [11]. It is well established that cerebral structure and function are altered in AD. Patients with AD show a reduced basal forebrain volume and hippocampus atrophy compared with the healthy controls [38]. Thus, the significant reduction in performance in the pace domain may be because of the structural and functional brain changes associated with pathological processes in AD. 

The variability domain was significantly associated with AD only in the dual-task condition. After adjusting for age, sex and education levels, a 1-point increase in the variability factor (counting backwards, by 175%) was associated with the risk of developing AD. In this study, the difference in outcomes between the dual-task and the single-task, in part, may be due to differences in the nature of these two tasks. The dual-task paradigm reflects the relationship between cognition and gait more sensitively when compared with the single-task paradigm [11]. Originally, Lundin-Olsson et al. noticed that some frail elderly patients stopped walking when they started a conversation with a walking companion, and he proposed that the inability to maintain a conversation while walking is a marker for future falls in older adults [39]. After that, observing people walking while they perform a secondary task (“dual-task paradigm”) has become a common way to assess the interaction between cognition and gait. The dual-task paradigm affords the opportunity to manipulate attention demands. There is a conflict between tasks during the performance of a dual task, as the two tasks interfere with each other and compete for the same brain resources [35]. Furthermore, a study suggested that the variability domain characterized by stride length variability may be a more sensitive predictor for identifying future cognitive decline than other gait domains [12]. The increase in gait parameters’ variability was found to be a loss of automaticity, and these people were more likely to fall [15]. Healthy people can walk smoothly while executing the dual-task, while people with AD are not capable of fluid movement when focusing on both tasks simultaneously. This may be due to the fact that paying attention to cognitive tasks can generate a greater cognitive load for participants with AD, which can interfere with the automaticity of walking movement. Thus, the increase in the variability domain was significantly associated with AD when all participants were asked to walk while counting backward.

There are unique signatures of gait impairments in different dementia disease subtypes, such as AD, Lewy body disease (LBD) and Vascular dementia (VD). The LBD group demonstrated greater impairments in asymmetry and variability compared with AD; both groups were more impaired in pace and variability domains than controls [40]. When compared to subjects with AD, subjects with vascular dementia walked more slowly and had a reduced step length [41]. In this study, the Tinetti scale was used, and 79 percent of patients with vascular dementia exhibited gait and balance disorders, compared with 25 percent of patients with AD. The rate of decline in mobility also differs, depending on the dementia subtype and rate of progression. Therefore, in addition to indicating the presence of dementia, gait analysis may have potential to distinguish disease subtypes.

We should admit that this cross-sectional study is preliminary and has some limitations. Firstly, this study was a cross-sectional study; therefore, we cannot evaluate the dynamic associations between gait domains and disease progression of AD. Secondly, the diagnosis of AD in our study was based on MRI, clinical symptoms and MMSE scores. And the majority of AD participants did not undergo the PET neuroimaging or CSF assays to make a definite diagnosis. Thirdly, the associations between the specific sub-items of MMSE and gait domains were not included in our study due to the limited study sample size. Fourthly, in our study, upper limb movements were not assessed, and it was impossible to determine the effect of upper limb movements on gait parameters. Finally, most of the participants with AD had taken medications, including antidepressants and antipsychotics, etc. In contrast, the HC population had not taken medication. As a result, the two groups were incomparable. Due to the unbalanced data, it is not possible to eliminate the influence of medication history on the gait of participants with AD. Consequently, further work is required to enhance the sample size and refine the selection criteria for participation, as well as the study methodology, to bolster the reliability of the results.

Despite these limitations, there are some strengths in our study. Our study explored the association between gait domains and AD under single and dual-task gait assessments, as well as to find a new, objective and simple gait domain to recognize the AD patients from healthy old people in clinical practice. In two gait assessments, the components of three gait domains (rhythm, pace and variability domains) were found to be the same. And our study found that the pace factor was independently associated with AD both in the free walk and counting backwards assessments and the variability factor was independently associated with AD only in the counting backwards assessment. Gait domains could help clinicians to fully comprehend the gait performance in AD patients and help clinicians to choose the appropriate combination of gait parameters to monitor the disease and its progression. Therefore, the pace and variability domains could be objective and simple markers for recognizing AD in clinical practice.

## 5. Conclusions

Our study found that the pace factor was independently associated with AD both in single- and dual-task gait assessments and the variability factor was independently associated with AD only in the dual-task gait assessment. Gait domains, objective and simple markers, could help clinicians fully comprehend the gait performance in AD patients and choose the appropriate combination of gait parameters to monitor the disease and its progression.

## Figures and Tables

**Figure 1 brainsci-13-01599-f001:**
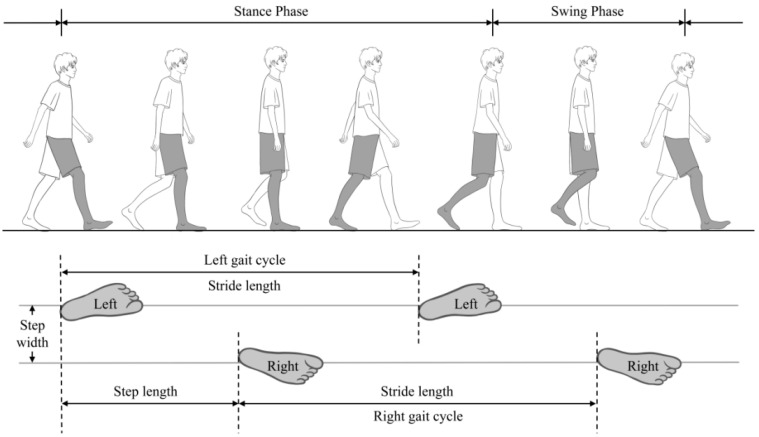
Diagram of the gait cycle.

**Figure 2 brainsci-13-01599-f002:**
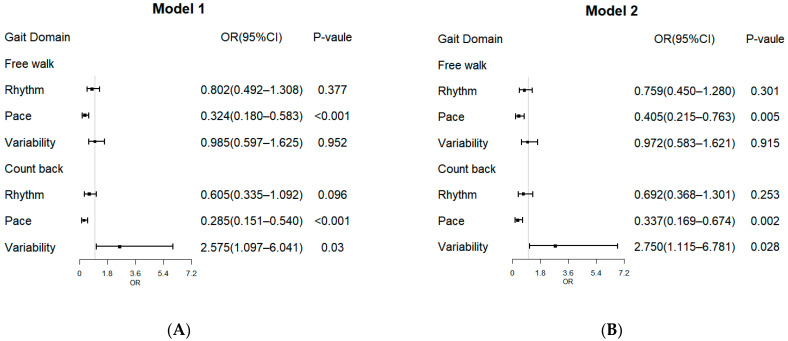
The forest plot of regression between AD and gait domain for two gait assessments. **A** Model 1. **B** Model 2. Model 1: unadjusted; Model 2: adjusted for age, sex, education levels. OR, odds ratio; CI, confidence interval.

**Table 1 brainsci-13-01599-t001:** Demographic and gait characteristics of AD participants and HC participants.

	AD (*n* = 41)	HC (*n* = 41)	*p*-Value
Demographic characteristics			
Age (years), mean ± SD	68.2 ± 8.1	62.1 ± 8.3	<0.001
Sex, female, *n* (%)	26 (63.4%)	20 (48.8%)	0.182
BMI (kg/m^2^), mean ± SD	23.2 ± 2.8	23.5 ± 3.3	0.655
Education levels, *n* (%)			0.010
Illiteracy	12 (29.3%)	4 (9.8%)	
Primary school	14 (34.1%)	27 (65.8%)	
Middle school and higher	15 (36.6%)	10 (24.4%)	
MMSE score, median (IQR)	13.0 (7.0–19.0)	25.0 (25.0–28.0)	<0.001
Medications, *n* (%)			<0.001
Aricept	38 (92.7%)	0 (0.0%)	
Memantine	25 (61.0%)	0 (0.0%)	
SSRI (sertraline)	5 (12.2%)	0 (0.0%)	
Antipsychotics			
Olanzapine	5 (12.2%)	0 (0.0%)	
Risperidone	2 (4.9%)	0 (0.0%)	
Quetiapine	1 (2.4%)	0 (0.0%)	
Cozapine	1 (2.4%)	0 (0.0%)	
Gait parameters			
Free walk			
Stride length (m), mean ± SD	0.94 ± 0.19	1.11 ± 0.17	<0.001
Gait velocity (m/s), mean ± SD	0.78 ± 0.20	0.93 ± 0.19	0.001
Gait frequency (steps/min), mean ± SD	98.50 ± 12.48	100.25 ± 10.26	0.489
Stance phase (%), mean ± SD	66.55 ± 3.18	64.57 ± 2.01	0.001
Swing phase (%), mean ± SD	33.45 ± 3.18	35.43 ± 2.02	0.001
Stride time (s), median (IQR)	1.19 (1.12–1.29)	1.17 (1.11–1.30)	0.633
Swing time (s), median (IQR)	0.78 (0.73–0.85)	0.75 (0.70–0.85)	0.294
Stride time variability (CV), median (IQR)	0.03 (0.02–0.04)	0.03 (0.02–0.04)	0.466
Swing phase variability (CV), median (IQR)	0.04 (0.03–0.05)	0.03 (0.03–0.04)	0.537
Count backward			
Stride length (m), mean ± SD	0.90 ± 0.18	1.10 ± 0.20	<0.001
Gait velocity (m/s), mean ± SD	0.63 ± 0.20	0.82 ± 0.20	<0.001
Gait frequency (steps/min), mean ± SD	82.98 ±17.16	89.56 ± 14.81	0.068
Stance phase (%), mean ± SD	68.93 ± 4.67	65.66 ± 2.76	<0.001
Swing phase (%), mean ± SD	31.07 ±4.67	34.34 ± 2.76	<0.001
Stride time (s), mean ± SD	1.52 ± 0.39	1.38 ± 0.26	0.058
Swing time (s), mean ± SD	0.46 ± 0.05	0.47 ± 0.06	0.399
Stride time variability (CV), median (IQR)	0.06 (0.03–0.11)	0.04 (0.03–0.06)	0.023
Swing phase variability (CV), median (IQR)	0.05 (0.04–0.08)	0.04 (0.03–0.05)	0.004

Note: AD, Alzheimer’s disease; HC, healthy controls; BMI, body mass index; MMSE, Mini-mental State Examination; CV, coefficient of variation.

**Table 2 brainsci-13-01599-t002:** Multiple logistic regression of gait features and AD.

Gait Parameters	OR	95% CI	*p*-Value
Free walk			
Stride length (m)	0.012	0.001–0.277	0.006
Gait velocity (m/s)	0.034	0.002–0.536	0.016
Gait frequency (steps/min)	0.980	0.938–1.024	0.375
Stance phase (%)	1.272	1.032–1.568	0.024
Swing phase (%)	0.786	0.638–0.969	0.024
Stride time (s)	6.166	0.368–103.435	0.206
Swing time (s)	8.363	0.399–175.135	0.171
Stride time variability (CV)	1.027	0.909–1.159	0.670
Swing phase variability (CV)	1.029	0.911–1.162	0.646
Count backward			
Stride length (m)	0.009	0.000–0.199	0.003
Gait velocity (m/s)	0.019	0.001–0.333	0.007
Gait frequency (steps/min)	0.984	0.952–1.016	0.327
Stance phase (%)	1.244	1.045–1.480	0.014
Swing phase (%)	0.804	0.676–0.957	0.014
Stride time (s)	2.421	0.435–13.480	0.313
Swing time (s)	0.021	0.000–136.087	0.389
Stride time variability (CV)	1.146	1.012–1.298	0.031
Swing phase variability (CV)	1.156	1.007–1.327	0.040

Notes: OR values were adjusted for gender, age and education. AD, Alzheimer’s disease; OR, odds ratio; CI, confidence interval; CV, coefficient of variation.

**Table 3 brainsci-13-01599-t003:** Factor loading of seven quantitative variables on three independent gait factors rotated and extracted via factor analysis.

Gait Parameters	Gait Domains		
	Rhythm Factor	Pace Factor	Variability Factor
Free walk			
Stride length (m)	0.148	**0.984**	−0.055
Gait velocity (m/s)	0.539	**0.812**	−0.083
Gait frequency (steps/min)	**0.934**	0.114	−0.18
Stance phase (%)	**−0.763**	−0.512	0.285
Swing phase (%)	**0.763**	0.512	−0.285
Stride time variability (CV)	−0.124	−0.055	**0.971**
Swing phase variability (CV)	−0.29	−0.108	**0.919**
Variance explained, %	34.98	31.13	28.46
Count backward			
Stride length (m)	0.179	**0.981**	−0.027
Gait velocity (m/s)	0.632	**0.710**	−0.190
Gait frequency (steps/min)	**0.903**	0.046	−0.339
Stance phase (%)	**−0.823**	−0.466	0.249
Swing phase (%)	**0.823**	0.466	−0.249
Stride time variability (CV)	−0.168	0.003	**0.960**
Swing phase variability (CV)	−0.432	−0.218	**0.821**
Variance explained, %	40.23	27.86	26.73

Notes: The highest loading features are shown in bold. CV, coefficient of variation.

## Data Availability

The data sets analyzed during the current study are available from the corresponding author on reasonable request.

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
