# Peer review of "Gait Domains May Be Used as an Auxiliary Diagnostic Index for Alzheimer’s Disease"

_brainsci, 2023, doi:10.3390/brainsci13111599_

Round 1
Reviewer 1 Report
Comments and Suggestions for Authors
Manuscript shows the potential gait alterations in Alzheimer patients and healthy controls
Good introduction, good synthesis and explanation of the subject of study.
Clear methodology with sample and criteria exposed; however, the procedure should be better explained. How many researchers were part of the study? Were the evaluators blinded? How many assessments were performed? On the same day? How long was the recruitment of data?
Were there any inclusion/exclusion criteria regarding age? At least to establish homogeneous groups. Which were the inclusion and exclusion criteria for healthy controls?
The three factors of the gait must be explained before to contextualize your results.
Is there any other neurological condition, such another dementia, that could be mention at discussion section in relation to this topic?
Some little mistakes to correct:
Line 71. Add a full stop after "al". Line 118 use capital letter after the full stop and delete the capital letter of "We" in line 122. Line 259 has two full stops together.
Author Response
Manuscript shows the potential gait alterations in Alzheimer patients and healthy controls
- Good introduction, good synthesis and explanation of the subject of study.
Response: We appreciate the reviewer’s positive evaluation of our work.
- Clear methodology with sample and criteria exposed; however, the procedure should be better explained. How many researchers were part of the study? Were the evaluators blinded? How many assessments were performed? On the same day? How long was the recruitment of data?
Response:We are grateful for the suggestion. To be more clear and in accordance with the reviewer concerns, we have added a brief description as follows.
A total of seven major principal researchers participated in this study, the details are as followed. Qi Duan: data curation, methodology, formal analysis, writing—original draft; Yinuo Zhang: data curation, methodology, writing—original draft; Weihao Zhuang and Wenlong Li: methodology, assistance of data curation; Jincai He and Zhen Wang: trained neurologists, responsible for the diagnosis of Alzheimer's disease; Haoran Cheng: supervision, Conceptualization, writing—review and editing.
The evaluators were performed by experienced neurologists who were blinded to the participants’ medical information. Participants were recruited from October 2019 to November 2021 and each participant was performed two walk trials including free walk and counting backwards on the same day. The detailed explanation is as follows: All participants with AD were patients attending the Memory Clinic of the Department of Neurology at the First Affiliated Hospital of Wenzhou Medical University, and HC participants were recruited from the Health Examination Center, all of whom were assessed and diagnosed by trained neurologists on the basis of clinical symptoms, MMSE scores and resonance imaging (MRI) prior to gait assessment. Gait assessment was subsequently performed at the same time of day. All participants performed two gait assessments including free walk and counting backwards. The two gait assessments were performed in order. We carried out the walk trials by free walk followed by counting backwards.
Recruitment of the data was continued from October 2019 to November 2021. And We have added this information to the revised manuscript. Please see the revised manuscript(page 3, lines 110–111).
- Were there any inclusion/exclusion criteria regarding age? At least to establish homogeneous groups.Which were the inclusion and exclusion criteria for healthy controls?
Response:Thanks for the comments.
In our study, there were fewer healthy control (HC) participants over 70 years of age recruited from the health screening center. Considering the limitation of sample size, we did not match the age between the AD participants and HC participants. No age homogeneous group was created is our limitation in this study. As we did consider age is an important factor affecting the gait, we had adjusted age factor in multivariate logistic regression to minimize the influence. We reported observed coefficient, significance and 95%CI. Further work is required to enhance the sample size to establish homogeneous groups to bolster the reliability of the results.
We have revised the text to address your concerns and hope that it is now clearer. Please see the revised manuscript (page 3, lines 133–135).
“HC participants were enrolled by the above inclusion and exclusion criteria. And HC participants had intact cognitive function, who did not meet the criteria of dementia or MCI.”
- The three factors of the gait must be explained before to contextualize your results.
Response:Thanks for the comments. We have revised the text to address your concerns.
For this information we refer to the article of Hollman et al. (reference 18 in the manuscript). This study obtained five gait domains through principal components factor analysis: : a “rhythm” domain was characterized by cadence and temporal parameters such as stride time; a ”phase” domain was characterized by temporariness parameters that constitute distinct divisions of the gait cycle; a ”variability” domain encompassed gait cycle and step variability parameters; a “pace” domain was characterized by parameters that included gait speed, step length and stride length; and a ”base of support” domain was characterized by step width and step width variability.
In our study, Factor analysis with varimax rotation yielded exactly three orthogonal factors. Consistent with the previous research, the identified gait domains were derived from the similar gait parameters: rhythm, pace, variability. We have included in the explanation regarding the three factors of the gait. (page 7, lines 237–246)
- Is there any other neurological condition, such another dementia, that could be mention at discussion section in relation to this topic?
Response:Thank you very much for pointing out this important issue. We agree with your suggestion. Thus, we have added the discussion about other types of dementia,such as vascular dementia. Please see the revised manuscript, discussion section (page 10, lines 322-332).
“There are unique signatures of gait impairments in different dementia disease subtypes , such as AD, Lewy body disease (LBD) and Vascular dementia (VD). The LBD group demonstrated greater impairments in asymmetry and variability compared with AD; both groups were more impaired in pace and variability domains than controls [39]. When compared to subjects with AD, subjects with vascular dementia walked more slowly and had a reduced step length [40]. In this study, the Tinetti scale was used, and 79 percent of patients with vascular dementia exhibited gait and balance disorders, compared with 25 percent of patients with AD. The rate of decline of mobility also differs, depending on the dementia subtype and rate of progression. Therefore, in addition to indicating the presence of dementia, gait analysis may have potential to distinguish disease subtypes.”
- Mc,Ardle.R.; Galna, B.; Donaghy, P.; Thomas, A.; Rochester, Do Alzheimer's and Lewy body disease have discrete pathological signatures of gait? Alzheimers Dement. 2019, 15, 1367-1377.
- Allan, L.M.; Ballard, C.G.; Burn, D.J.; Kenny, R.A. Prevalence and severity of gait disorders in Alzheimer's and non-Alzheimer's dementias. J. Am. Geriatr. Soc. 2005, 53, 1681-7.
- Some little mistakes to correct:
Line 71. Add a full stop after "al". Line 118 use capital letter after the full stop and delete the capital letter of "We" in line 122. Line 259 has two full stops together.
Response:Thanks for your careful checks. We are sorry for our carelessness. Based on your comments, we have made changes accordingly.

Reviewer 2 Report
Comments and Suggestions for Authors
The main purpose of the work was to use factor analysis to explore the association between gait domains and Alzheimer's disease
The paper is generally well written , the results well presented.
The work is written following the steps of the scientific method.
With the development of sensor technology and gait data analyzing techniques, gait analysis using wearable sensors has become a widespread and useful tool for both clinical practice and biomechanical research.
Introduction
The introduction of the study is well structured, the rationale behind the study is written in a clear and understandable way
Materials and Methods
This section contains enough information to understand and possibly repeat the study. Almost every aspect of the study has been considered and explained in detail but some technical detail of instrumed used (JiBuEn®) missing, such as the stability of sensor signals, the reliability and validity of analytical algorithms for kinematics and kinetics gait analysis.
Results and discussion
The results presented are complete, the length and content of the discussion communicate the main information of the paper.
The study is well designed. however I have a comment I’d like to express: the authors provided limitations for this study and future potential studies related to the topic necessary to further clarify the matter: in particular add a chapter “limitations and future direction” is important to explain the limits above all from a clinical point of view:
Gait is the result of many forces and derives from the coordination of muscles’ movement and balance machinery, requiring integrity of central and peripheral neurophysiological mechanisms. It would be useful to address the end-points to study also in terms of upper extremities , why upper limb movements were not assessed? why postural sway were not assessed?
Please explain why a 10 m cut-off was used, and the actual experimental data so on. Consider that functional tests do require to walk a specific distance. Additionally, do sensors record data during a specific test? Is there a problem analysing recordings, dissecting phases etc?
It is feasible not to include other neurological diseases control data? Please consider the lack of such type of data within the experimental work, to understand both their purpose and their (additive) value in gait recording using wearable sensors. Are they established deviations from the AD to control population regarding clearance, gait speed, stride velocity, length, cadence, step duration, extension, asymmetry?
Comments on the Quality of English LanguageModerate editing of English language required
Author Response
The main purpose of the work was to use factor analysis to explore the association between gait domains and Alzheimer's disease
The paper is generally well written , the results well presented.
The work is written following the steps of the scientific method.
With the development of sensor technology and gait data analyzing techniques, gait analysis using wearable sensors has become a widespread and useful tool for both clinical practice and biomechanical research.
- Introduction
The introduction of the study is well structured, the rationale behind the study is written in a clear and understandable way.
Response: We appreciate the reviewer’s positive evaluation of our work.
- Materials and Methods
This section contains enough information to understand and possibly repeat the study. Almost every aspect of the study has been considered and explained in detail but some technical detail of instrumed used (JiBuEn®) missing, such as the stability of sensor signals, the reliability and validity of analytical algorithms for kinematics and kinetics gait analysis.
Response: We deeply appreciate the reviewer’s suggestion. According to the reviewer’s comment, we have added a more detailed interpretation regarding JiBuEn® and added to the relevant literature. Please see the revised manuscript (page 4, lines 152–166).
“Gait performance was measured by JiBuEn® gait analysis system (Hangzhou Zhihui health management co., LTD) consisting of five inertial sensors and a pair of shoes with gyroscope and 32 pressure sensors [30]. Five inertial sensors are placed to subject’s waist, thighs and calves by nylon straps. Signals from the sensors are sampled and transferred through Bluetooth and received by a receiver connected to computer. JiBuEn® system is one of portable wearable devices that have been developed and used in measuring gait with low cost, simple implementation, and even instant report [31]. Detailed experimental design, algorithm for gait parameters and validated method were reported in previous study [32], and systematically evaluated the validity of JiBuEn® [33]. The high-order low-pass filter and hexahedral calibration technique were employed in data pre-processing, which reduces high-frequency noise interference and installation errors produced by sensor devices. Moreover, accumulative errors were also corrected based on the zero-correction algorithm. The final gait parameters were obtained by fusing acceleration data and posture, which is calculated using quaternary complementary filtering technique.”
- Phanpho, C., Rao, S., Moffat, M.: Immediate effect of visual, auditory and combined feedback on foot strike pattern. Gait Posture 74, 212–217 (2019)Return to ref 11 in article.
- Tao, S.; Zhang, X. W.; Cai, H.Y.; Lv, Z.P.; Hu, C.Y.; and Xie, H.Q. Gait based biometric personal authentication by using MEMS inertial sensors. J. Ambient Intell. Humaniz. Comput. 2018 9, 1705–1712.
- Gao, Q.; Lv, Z.; Zhang, X.; Hou, Y.; Liu, H.; Gao, W.; Chang, M.; Tao, S. Validation of the JiBuEn® System in Measuring Gait Parameters. International. Conference. On. Human. Interaction. And. Emerging. Technologies. 2021.
- Results and discussion
The results presented are complete, the length and content of the discussion communicate the main information of the paper.
The study is well designed. however I have a comment I’d like to express: the authors provided limitations for this study and future potential studies related to the topic necessary to further clarify the matter: in particular add a chapter “limitations and future direction” is important to explain the limits above all from a clinical point of view:
Gait is the result of many forces and derives from the coordination of muscles’ movement and balance machinery, requiring integrity of central and peripheral neurophysiological mechanisms. It would be useful to address the end-points to study also in terms of upper extremities , why upper limb movements were not assessed? why postural sway were not assessed?
Response:Thanks for the comments. We fully agree with your view above.
Your suggestion provides a direction for our next research and We have added this deficiency to the limitation section. Please see the revised manuscript (page 10, lines 340–341).
“Fourthly, in our study, upper limb movements were not assessed and it was impossible to determine the effect of upper limb movements on gait parameters.”
We have also done some soul-searching on this point and reviewed the relevant information. Gait is the action of walking (locomotion). It is a complex, whole-body movement, that requires the coordinated action of many joints and muscles of our musculoskeletal system. It mostly includes the movements of the lower limbs, upper limbs, pelvis and spine. During normal walking, humans swing their upper limbs alternately, each upper limb swinging in phase with contra lateral lower. Arm and leg movements are linked during locomotion and a defined frequency relationship between arm and leg movements is present (Wannier, T.; Bastiaanse, C.; Colombo, G.; Dietz V. Arm to leg coordination in humans during walking, creeping and swimming activities. Exp. Brain. Res. 2001, 14, 375-9). Therefore, assessing upper limb movements is crucial.
However, The main purpose of this study is to to find a new, objective and simple gait domain to recognize the AD patients from healthy old people in clinical practice. And, we sought to contribute to promotion of the wearable intelligent sensors in the elderly population. This system (JiBuEn®) used in measuring gait with low cost, simple implementation, and even instant report which can be better help monitor the disease and its progression. The analysis system we used mainly evaluates gait parameters for lower limb movements. Because it is consisted of five inertial sensors and a pair of shoes with gyroscope and 32 pressure sensors. Thus, we mainly focus on gait parameters of lower limb movements.
In the future, We will do further work to improve the assessment system, enhance the sample size and refine the selection criteria for participation, as well as the study methodology, to quantify not only lower limb but also upper limb movement during gait and help clinicians choose the appropriate the combination of gait parameters to monitor the disease and its progression.
Among the gait parameters in our study were included Swing time and Swing phase variability (CV). Please see page 7 line 252: AD participants had larger variability of swing phase (P = 0.004; Table 1) which were only found in counting backwards. Page 7 line 253: arger variability of stride time (P = 0.031; Table 2) and swing phase (P = 0.040; Table 2) were significantly associated with AD, which were only found in counting backwards. Page 7 line 256: the third factor was named variability factor, loading heavily on stride time variability and swing phase variability.
- Please explain why a 10 m cut-off was used, and the actual experimental data so on. Consider that functional tests do require to walk a specific distance. Additionally, do sensors record data during a specific test? Is there a problem analysing recordings, dissecting phases etc?
Response:Thanks for the comments. I have made the following explanation about this issue, and I have listed the relevant references below.
In 1987 Wade et al first described and documented the specific use of a ten-metre walking test to monitor recovery of gait following stroke [1]. This involved timing how long it takes subjects to walk ten metres from a standing start, moving at their usual speed with their usual walking aids. Though spec-ific discussion as to why this distance has been chosen is difficult to find, its practicality and validity is understandable. Ten metres is probably the minimum functionally significant distance in the recovery of independent walking. It is also probably a typical distance in clinical gait remediation, in terms of the free length of treatment areas and/or parallel walking bars. Ten metres is thus both a practical and meaningful distance to use. Nowaday, The 10-Meter Walk Test is a commonly used tool for assessing gait speed in individuals with gait limitations and this test has been validated in various diseases and populations, such as stroke [2,3], spinal cord injury [4], Parkinson's disease [5], Alzheimer's disease [6]. Therefore in this study we also used a 10 m cut-off to assess the participants' gait parameters.
- Wade, D.T.; Wood, V.A.; Heller, A.; Maggs, J.; Langton Hewer, R. Walking after stroke. Measurement and recovery over the first 3 months. Scand. J. Rehabil. Med. 1987, 19, 25-30.
- Wade, D.T.; Collen, F.M.; Robb, G.F.; Warlow, C.P. Physiotherapy intervention late after stroke and mobility. BMJ. 1992, 304, 609-13.
- Flansbjer, U.B.; Holmback, A.M.; Downham, D.; Patten, C.; Lexell, J. Reliability of gait performance tests in men and women with hemiparesis after stroke. J. Rehabil. Med. 2005, 37 , 75-82 .
- Lam, T.; Noonan, V.K.; Eng, J.J. A systematic review of functional ambulation outcome measures in spinal cord injury. Spinal Cord. 2007, 46, 246-254 .
- Lindholm, B.; Nilsson, M.H.; Hansson, O.; Hagell, P. The clinical significance of 10-m walk test standardizations in Parkinson's disease. J. Neurol. 2018, 265, 1829-1835.
- Ansai, J.H.; Andrade, L.P.; Rossi, P.G.; Takahashi, A.C.M.; Vale, F.A;C.; Rebelatto, J.R. Gait, dual task and history of falls in elderly with preserved cognition, mild cognitive impairment, and mild Alzheimer's disease. Braz. J. Phys. Ther. 2017, 21, 144-151.
There will be no problems with analysing recordings, dissecting phases. This gait system (JiBuEn®) could record the detailed data of walking process in multi-dimensional; and automatically calculate gait parameters in real time, such as gait velocity, length, stride time, gait variability.
- It is feasible not to include other neurological diseases control data? Please consider the lack of such type of data within the experimental work, to understand both their purpose and their (additive) value in gait recording using wearable sensors. Are they established deviations from the AD to control population regarding clearance, gait speed, stride velocity, length, cadence, step duration, extension, asymmetry?
Response: Thanks for your comments, I will give the following explanation for this issue.
Our study focuses on the relationship between the gait parameters and AD. In order to improve the reliability of the results, other neurological disorders were excluded as confounders in the study design, such as stroke, Parkinson’s disease, Huntington's disease or myasthenia gravis. There are many previous studies that have shown that Parkinson's disease, stroke and other diseases are also related to gait. Hence the AD participants and HC participants in our study were free of other neurological disorders.

Reviewer 3 Report
Comments and Suggestions for Authors
The manuscript is very well written. The results section raises a few concerns although most of those concerns are already well discussed with clear explanations in the discussion section of the manuscript.
I have the following comments:
1. Tables are great although, for the readers, if possible, plots should be added.
2. Authors should mention the relation between the variability in the age of participants vs the variability in the gait parameters.
-
The age range for AD subjects was 60-74 years whereas the age range was 54-70 years for healthy controls in the study, gait in healthy controls could be different due to age difference.
-
Authors may compare the age effect. In line 228 - what do the authors mean by adjusting for age? How was the age factor adjusted? Please describe. Similarly, describe how were the adjustments made for sex and education level.
3. Provide a reference for multivariate regression analysis.
4. In the discussion, authors use the word predicted, e.g. line 236 - authors mention the pace factor predicting AD. Since the experiments were done with 60-74-year-old AD subjects i.e. people who already had AD, the word 'predicted' seems misleading without the necessary details about the usage of the term “prediction” and specifying how accurate is the prediction.
-
Authors should specify (with references) that the word prediction refers to the algorithmic prediction of the test dataset after training the algorithm on the training dataset if that is the case.
-
The authors should clarify the percentage of data used for training and testing the algorithm.
-
Also, authors should provide ROC curve plots and quantitative parameters that specify the accuracy, precision, specificity, and sensitivity of the algorithm including the estimates of the true and false, positive and negative for the dataset and the logistic regression model used.
-
Authors may provide the Confusion Matrix and Classification Error Rate information for their dataset and model used.
Minor checks needed
Author Response
The manuscript is very well written. The results section raises a few concerns although most of those concerns are already well discussed with clear explanations in the discussion section of the manuscript.
I have the following comments:
- Tables are great although, for the readers, if possible, plots should be added.
Response:We have improved it according to the Reviewer’s comments. A forest plot has been developed to show the results of logistic regression, as this gives the reader a much better understanding of the data. We have modified “Table 4.” to “Figure 2.”
- Authors should mention the relation between the variability in the age of participants vs the variability in the gait parameters.
The age range for AD subjects was 60-74 years whereas the age range was 54-70 years for healthy controls in the study, gait in healthy controls could be different due to age difference.
Authors may compare the age effect. In line 228 - what do the authors mean by adjusting for age? How was the age factor adjusted? Please describe. Similarly, describe how were the adjustments made for sex and education level.
Response:Thanks for the comments. We have made the following explanation about this issue. Because of our small sample size, we didn’t show the data stratified by age. In the future, further work is required to enhance the sample size to establish homogeneous groups to bolster the reliability of the results.
However, as we did consider age is an important factor affecting the gait, and we adjusted age factor in multivariate logistic regression to minimize the influence. We reported observed coefficient, significance and 95%CI. There is also possible that sex and educational level are potential confounders. So we simultaneously adjusted for age, sex, and education level through multivariate logistic regression.
- Provide a reference for multivariate regression analysis.
Response: Thanks for the comments. We have revised the text to address your concerns and hope that it is now clearer. we have added the reference for multivariate regression analysis (reference 35).
- In the discussion, authors use the word predicted, e.g. line 236 - authors mention the pace factor predicting AD. Since the experiments were done with 60-74-year-old AD subjects i.e. people who already had AD, the word 'predicted' seems misleading without the necessary details about the usage of the term “prediction” and specifying how accurate is the prediction.
Authors should specify (with references) that the word prediction refers to the algorithmic prediction of the test dataset after training the algorithm on the training dataset if that is the case. The authors should clarify the percentage of data used for training and testing the algorithm.
Also, authors should provide ROC curve plots and quantitative parameters that specify the accuracy, precision, specificity, and sensitivity of the algorithm including the estimates of the true and false, positive and negative for the dataset and the logistic regression model used.
Authors may provide the Confusion Matrix and Classification Error Rate information for their dataset and model used.
Response:Thank you for the above suggestion. We fully agree with your view above that the use of the word "predicted" is not rigorous enough. Because clinical prediction models require a sufficient sample size, but the sample size included in the analysis in our study was only 82, which is not sufficient to divide this dataset into the training set and test set. Therefore we changed the term “predicted” to ”was associated with”. Please see the revised manuscript (page 9, lines 288 and 300).
“After adjusting for age, sex and education levels, a 1-point decline in pace factor (free walk, by 59.5%; counting backwards, by 66.3%) was associated with risk of developing AD.” “Variability domain was significantly associated with AD only in dual-task condition. After adjusting for age, sex and education levels, a 1-point increase in variability factor (counting backwards, by 175%) was associated with risk of developing AD.”
The ROC curve plots was used to assess the discriminative performance of the Logic regression model. The ROC curve of pace factor in the free walk trials showed an AUC of 0.767 (95% CI: 0.662-0.871) (figure A).Then, in the Count backward trials, pace factor and variability factor showed the AUC of 0.760 (95% CI: 0.654-0.864) and 0.602 (95% CI: 0.476-0.728), respectively (figure B, C). Due to small sample size, the ROC seems not to be a good statistics for sensitivity and specificity. Thus, We did not add this result in the manuscript.
In the following study, further work is required to enhance the sample size and refine the selection criteria for participation, as well as the study methodology, to bolster the reliability of the results.

Reviewer 4 Report
Comments and Suggestions for Authors
>Please, reorganize the abstract.
>The keywords should be unified.
>The introduction should include additional information about applications of the artificial intelligence for the medical data processing.
>The main tools used for the data analysis are: the Principal Component Analysis and the logistic regression. Some details related to those techniques should be mentioned.
>Have you used data pre-processing (e.g. scaling)?
>The short description of the article structure and the novelty highlights are commonly presented (in scientific papers) in the final part of the introduction.
>The section 2.4. contains short information about the numerical calculations. However, following details related to the algorithms implementation are needed.
>The double dot is in the text (line 259).
>Have you collected (section 2.3.) the enough number of data (for fair results analysis)?
>How were the parameters of the logistic regression model calculated?
>Have you analyzed external unexpected disturbance in the analyzed data?
Comments on the Quality of English Language
It is acceptable, minor improvements of the English grammar are required.
Author Response
Response to reviewer #4:
- Please, reorganize the abstract.
Response:Thank you for the above suggestion. I've reorganized the abstract. Please see the revised manuscript (page 1, lines 12–25).
“Background: Alzheimer's disease (AD) is a progressive neurodegenerative disorder with cognitive dysfunction and behavioral impairment. We aimed to use Principal components factor analysis to explore the association between gait domains and AD under single and dual-task gait assessments. Methods: 41 AD participants and 41 healthy controls (HC) were enrolled in our study. Gait parameters was measured by JiBuEn® gait analysis system. The principal component method was used to conduct an orthogonal maximum variance rotation factor analysis of quantitative gait parameters. Multiple logistic regression was used to adjust for potential confounding or risk factors. Results: Based on factor analysis, three domains of gait performance were identified both in free walk and counting backward assessments: “rhythm” domain, “pace” domain and “variability” domain. Compared with HC, we found that pace factor was independently associated with AD both in two gait assessments; variability factor was independently associated with AD only in counting backwards assessment; and statistic difference still remained after adjusting for age, sex and education levels. Conclusions: Our findings indicate that gait domains may be used as an auxiliary diagnostic index for Alzheimer's disease.”
- The keywords should be unified.
Response:Thanks for your careful checks. We are sorry for our carelessness. Based on your comments, we have made changes accordingly. Please see the revised manuscript (page 1, lines 26–27).
“Keywords: Alzheimer's disease; Gait domain; Principal components factor analysis; Dual-task gait assessments”
- The introduction should include additional information about applications of the artificial intelligence for the medical data processing.
Response:Thanks for the comments. We apologize that the introduction about our study here was insufficient at first. (page 2, lines 68–80)
“Artificial intelligence (AI) has led to numerous technical innovations in medicine and revolutionized the conventional mode of medicine, especially neurological diseases. For example, using the method of human-computer interaction for early warning and ancillary diagnosis of nervous system diseases [14]. The brainecomputer interface (BCI) is the linkage of the brain to computers through scalp, subdural or intracortical electrodes to improve control of movement disorders and memory enhancement [15]. Sensor technology is the most basic accessory of artificial intelligence. Wearable intelligent sensors are inexpensive, convenient, and efficient, which has made them one of the most popular types of electrochemical sensors. Meanwhile as a clinical tool applied in the rehabilitation and diagnosis of medical conditions and sport activities, gait analysis using wearable sensors shows great prospects [16]. By means of this technology, several studies have demonstrated that the daily poor gait performance was associated with the risk of falls [17], Parkinson’s disease [7], and AD [18].”
- Li, Y.; Wang, Liu, P.; Huang, J.; Fan, X.M.; Tian, F. Application of Human-Computer Interaction Technology in Ancillary Diagnosis of Nervous System Diseases: Current Situation and Prospect. Medical. Journal. Of. Peking. Union. Medical. College. Hospital. 2021, 12, 608-613.
- Rosenfeld,J.V.; Wong, Y. Neurobionics and the brain-computer interface: current applications and future horizons. Med. J. Aust. 2017, 206, 363-368.
- Tao, W.; Liu, T.; Zheng, R.; Feng, H. Gait analysis using wearable sensors. Sensors (Basel). 2012, 12, 2255-83.
- van Schooten, K.S.; Pijnappels, M.; Rispens, S.M.; Elders, P.J.; Lips, P.; van Dieen, J.H. Ambulatory fall-risk assessment: amount and quality of daily-life gait predict falls in older adults. J. Gerontol. A Biol. Sci. Med. Sci. 2015, 70, 608–615.
- Hsu, Y.L.; Chung, P.C.; Wang, W.H.; Pai, M.C.; Wang, C.Y.; Lin, C.W.; et al. Gait and balance analysis for patients with Alzheimer’s disease using an inertial-sensor-based wearable instrument. IEEE J. Biomed. Health Inform. 2014, 18, 1822–1830.
- The main tools used for the data analysis are: the Principal Component Analysis and the logistic regression. Some details related to those techniques should be mentioned.
Response:We appreciate your comment. In the revised manuscript, we have added more details on thestatistical methods , including the specific tests used and why they were chosen for our analysis.
Please see the revised manuscript (page 2, lines 92–98).
“Given that the gait parameters are closely correlated with each other and gait parameters alone may not fully explain the gait performance. To address this issue,we used factor analysis to identify independent gait domains derived from quantitative assessments. Principal components factor analysis organizes multiple observations into communalities that correlate with a lesser number of unobserved thematic constructs, thus allowing an investigator to partition a large number of parameters into a lesser number that characterize distinct domains of the parameters being measured [23].”
Please see the revised manuscript(page 5, lines 189–199).
“The initial factors were then subjected to an orthogonal varimax rotation to reduce the larger number of highly correlated variables to a smaller number of uncorrelated independent predictors to be used in the final analysis. Principal components factor analysis with varimax rotation was used to examine factors with eigenvalues exceeding 1.0 that characterized gait performance. Parameters with correlation loadings of 0.5 or higher were interpreted as being significant contributors to the factor. To assess the association between AD and gait domain for two gait assessments, the multivariate logistic regression model with a stepwise backward selection process was applied to adjust potential confounding or risk factors based on age, gender, and education level [35]. Meanwhile, a forest plot has been developed to show the results of logistic regression.”
- Have you used data pre-processing (e.g. scaling)?
Response:Thanks for the comments. We have made the following explanation about this issue.
Firstly, In our study, Gait performance was measured by JiBuEn® gait analysis system. This system can automatically pre-process the collected data. And we have added some technical detail to the revised manuscript. Please see the revised manuscript (page 4, lines 161–166).
“The high-order low-pass filter and hexahedral calibration technique were employed in data pre-processing, which reduces high-frequency noise interference and installation errors produced by sensor devices. Moreover, accumulative errors were also corrected based on the zero-correction algorithm. The final gait parameters were obtained by fusing acceleration data and posture, which is calculated using quaternary complementary filtering technique.”
Secondly, The main statistical method used in our study (Principal Component Analysis) is also a commonly used dimensionality reduction method in data processing that transforms multiple indicators into a few composite indicators.
- The short description of the article structure and the novelty highlights are commonly presented (in scientific papers) in the final part of the introduction.
Response:Thanks for the comments. We have made the following explanation about this issue.
Firstly, In our study, Gait performance was measured by JiBuEn® gait analysis system. This system can automatically pre-process the collected data. And we have added some technical detail to the revised manuscript. Please see the revised manuscript (page 4, lines 161–166).
“The high-order low-pass filter and hexahedral calibration technique were employed in data pre-processing, which reduces high-frequency noise interference and installation errors produced by sensor devices. Moreover, accumulative errors were also corrected based on the zero-correction algorithm. The final gait parameters were obtained by fusing acceleration data and posture, which is calculated using quaternary complementary filtering technique.”
Secondly, The main statistical method used in our study (Principal Component Analysis) is also a commonly used dimensionality reduction method in data processing that transforms multiple indicators into a few composite indicators.
- The section 2.4. contains short information about the numerical calculations. However, following details related to the algorithms implementation are needed.
Response:Thanks for the comments. We think this is an excellent suggestion. In the revised manuscript, we have added more details on related to the algorithms implementation in section 2.4. Please see the revised manuscript (page 5, lines 189–199).
“The initial factors were then subjected to an orthogonal varimax rotation to reduce the larger number of highly correlated variables to a smaller number of uncorrelated independent predictors to be used in the final analysis. Principal components factor analysis with varimax rotation was used to examine factors with eigenvalues exceeding 1.0 that characterized gait performance. Parameters with correlation loadings of 0.5 or higher were interpreted as being significant contributors to the factor. To assess the association between AD and gait domain for two gait assessments, a multivariate logistic regression model with a stepwise backward selection process was applied to adjust potential confounding or risk factors, adjusted for age, sex and education levels[35]. Meanwhile, a forest plot has been developed to show the results of logistic regression.”
- The double dot is in the text (line 259).
Response:Thanks for the comments. It is really a giant mistake to the whole quality of our article. We feel sorry for our carelessness. We have corrected it and we also feel great thanks for your point out.
- Have you collected (section 2.3.) the enough number of data (for fair results analysis)?
Response:Thanks for the comments. We have made the following explanation about this issue.
Ultimately, 82 participants (41 AD and 41 HC) were enrolled in our study. We collected the participants’ demographic and clinical data. The gait system could record the detailed data of walking process in multi-dimensional; and automatically calculate gait parameters in real time, such as gait velocity, length, stride time, gait variability. However, we followed the general rule for multivariate analysis, stating that there should be at least 15 cases per included predictor variable (Stevens, J. (1996). Applied multivariate statistics for the social sciences). In the employed multivariate analysis, we had 82 cases and 22 independent variables. Thus, We t consider the sample size as adequate for the current statistical analyses and seems sufficient to answer their primary research question. In the future, further work is required to enhance the sample size and refine the selection criteria for participation, as well as the study methodology, to bolster the reliability of the results.
- How were the parameters of the logistic regression model calculated?
Response:Thanks for the comments. I have already given the following answers to this question. To assess the association between AD and gait domain for two gait assessments, we used a multivariate logistic regression model with a stepwise backward selection process to adjust potential confounding or risk factors based on age, gender, and education level.
- Have you analyzed external unexpected disturbance in the analyzed data?
Response:Thanks for the comments. We have made the following explanation about this issue.
External unexpected disturbance may be associated with a number of aspects, such as the environment of assessment, researchers, participants. While designing the study, we have taken some measures to prevent this event from happening. Firstly, gait assessments were held in a quiet well-lit environment, virtually eliminating unintentional interference from the external environment. Secondly, participants are assisted by three experienced researchers during gait assessment to prevent accidents. Finally, participants with other irresistible unexpected interference were excluded from the study. Therefore, the gait parameter data we have obtained are true and reliable and have not been interfered with by external accidental factors.
We apologize that the exclusion criteria for participants were not complete, so we have reworked the text. Please see the revised manuscript (page 3, lines 124–125).
“The exclusion criteria were as follows: (a) dyskinesia or leg problems, such as knee replacements or hip replacements; (b) major central nervous system disease, such as stroke, Parkinson’s disease, Huntington's disease or myasthenia gravis; (c) major psychiatric disorders which may impair cognition and gait, such as schizophrenia, bipolar affective disorder or alcohol abuse; (d) Participants experienced unexpected interference during gait assessment; (e) severely impaired cognitive function or unable to understand and complete the three prescribed walking tests; and (f) unwilling to sign the informed consent.”
